# Oncogenic KRAS-Induced Protein Signature in the Tumor Secretome Identifies Laminin-C2 and Pentraxin-3 as Useful Biomarkers for the Early Diagnosis of Pancreatic Cancer

**DOI:** 10.3390/cancers14112653

**Published:** 2022-05-27

**Authors:** Mohammad Azhar Kamal, Imran Siddiqui, Cristina Belgiovine, Marialuisa Barbagallo, Valentina Paleari, Daniela Pistillo, Chiara Chiabrando, Silvia Schiarea, Barbara Bottazzi, Roberto Leone, Roberta Avigni, Roberta Migliore, Paola Spaggiari, Francesca Gavazzi, Giovanni Capretti, Federica Marchesi, Alberto Mantovani, Alessandro Zerbi, Paola Allavena

**Affiliations:** 1Department of Immunology, Humanitas Clinical and Research Center-IRCCS, 20089 Rozzano, Italy; mkamal@rcsi.com (M.A.K.); imrans@miltenyi.com (I.S.); c.belgiovine@smatteo.pv.it (C.B.); marialuisa.barbagallo@humanitasresearch.it (M.B.); barbara.bottazzi@humanitasresearch.it (B.B.); roberto.leone@humanitasresearch.it (R.L.); ravigni@agcbio.com (R.A.); migliore.roberta@hsr.it (R.M.); federica.marchesi@humanitasresearch.it (F.M.); alberto.mantovani@humanitasresearch.it (A.M.); 2Biobank, Humanitas Clinical and Research Center-IRCCS, 20089 Rozzano, Italy; valentina.paleari@cancercenter.humanitas.it (V.P.); daniela.pistillo@cancercenter.humanitas.it (D.P.); 3Department of Environmental Health Sciences, Istituto di Ricerche Farmacologiche Mario Negri-IRCCS, 20156 Milan, Italy; chiara.chiabrando@marionegri.it (C.C.); silvia.schiarea@marionegri.it (S.S.); 4Department of Pathology, Humanitas Clinical and Research Center-IRCCS, 20089 Rozzano, Italy; paola.spaggiari@humanitas.it; 5Pancreatic Surgery Unit, Humanitas Clinical and Research Center-IRCCS, 20089 Rozzano, Italy; francesca.gavazzi@cancercenter.humanitas.it (F.G.); giovanni.capretti@hunimed.eu (G.C.); alessandro.zerbi@hunimed.eu (A.Z.); 6Department of Biomedical Sciences, Humanitas University, 20072 Pieve Emanuele, Italy; 7Department of Medical Biotechnology and Translational Medicine, University of Milan, 20129 Milan, Italy; 8The William Harvey Research Institute, Queen Mary University of London, London EC1M 6BQ, UK

**Keywords:** pancreatic ductal adenocarcinoma, early diagnosis, biomarker, LAMC2, PTX3, proteomics, secretome, ELISA

## Abstract

**Simple Summary:**

Pancreatic ductal adenocarcinoma is an aggressive neoplasia that responds poorly to treatments and is frequently recognized at advanced stages. Early diagnosis and the possibility to undergo resective surgery would increase the rate of patient survival and the chance of a definitive cure. In search of candidate biomarkers to improve laboratory tests for early diagnosis, we have characterized the proteins secreted by pancreatic cells expressing the oncogene *KRAS^G12V^*. Of several upregulated proteins, the expression of seven proteins was quantified in tumors of surgically resected patients. For two of these proteins, Laminin-C2 and Pentraxin-3, the plasma levels were significantly higher in patients than in healthy donors, and their good laboratory performance makes them two promising biomarkers of pancreatic cancer.

**Abstract:**

*KRAS* mutations characterize pancreatic cell transformation from the earliest stages of carcinogenesis, and are present in >95% of pancreatic ductal adenocarcinoma (PDAC) cases. In search of novel biomarkers for the early diagnosis of PDAC, we identified the proteins secreted by the normal human pancreatic cell line (HPDE) recently transformed by inducing the overexpression of the *KRAS^G12V^* oncogene. We report a proteomic signature of *KRAS*-induced secreted proteins, which was confirmed in surgical tumor samples from resected PDAC patients. The putative diagnostic performance of three candidates, Laminin-C2 (LAMC2), Tenascin-C (TNC) and Pentraxin-3 (PTX3), was investigated by ELISA quantification in two cohorts of PDAC patients (*n* = 200) eligible for surgery. Circulating levels of LAMC2, TNC and PTX3 were significantly higher in PDAC patients compared to the healthy individuals (*p* < 0.0001). The Receiver Operating Characteristics (ROC) curve showed good sensitivity (1) and specificity (0.63 and 0.85) for LAMC2 and PTX3, respectively, but not for TNC, and patients with high levels of LAMC2 had significantly shorter overall survival (*p* = 0.0007). High levels of LAMC2 and PTX3 were detected at early stages (I–IIB) and in CA19-9-low PDAC patients. In conclusion, pancreatic tumors release LAMC2 and PTX3, which can be quantified in the systemic circulation, and may be useful in selecting patients for further diagnostic imaging.

## 1. Introduction

Pancreatic ductal adenocarcinoma (PDAC), the most common type of pancreatic cancer, is an aggressive neoplasia with very poor prognosis and a survival rate at 5 years of only 3–7%. PDAC incidence is relatively low, but is likely to rise in the next few years [1,2]. PDAC is intrinsically resistant to chemotherapy and lacks effectively targetable oncogenic drivers [3,4,5]. More than 80% of patients present with an advanced stage of the disease at diagnosis, due to an undetected clinical course [1,3,6]. PDAC is characterized by early invasion into adjacent tissues and metastasis formation that hampers the possibility for surgical resection; subjects effectively eligible for surgery constitute only about 15–20% of total PDAC patients. Earlier detection of ductal adenocarcinoma is key to increase the proportion of resectable patients, surgery being the only potentially curative treatment for improving the clinical outcome [7,8,9]. Following surgery and adjuvant therapy, patient survival at 5 years increases up to 30% [4,5,6,7]. Diagnosis of PDAC is routinely achieved with conventional imaging tools, including computerized tomography (CT) scanning, magnetic resonance imaging (MRI), endoscopic ultrasonography (EUS) and endoscopic retrograde cholangiopancreatography (ERCP); however, these procedures are invasive, costly and not appropriate for a large number of patients [1,7,10]. The ideal diagnostic biomarker for pancreatic cancer should be non-invasive, cost-effective and able to detect early cancers or high-risk lesions with a high degree of sensitivity and specificity. CA19-9 is the only serological tumor marker approved by the FDA and used routinely for the detection and monitoring of PDAC progression; it has 79–81% sensitivity and 82–90% specificity for the diagnosis of PDAC [11,12,13]. However, CA19-9 also gives false positive results in benign pancreaticobiliary diseases [14,15]. Several studies reported a number of non-invasive diagnostic and/or prognostic markers of pancreatic cancer that have been tested alone or in combination, such as: CA125, CEA, CEACAM1, MUC1, MIC1/GDF15, REG3A/PAP1, PKM2 and AXL, as well as auto-antibodies, fecal microbiome signatures and myeloid-derived suppressor cells [16,17,18,19,20,21,22,23,24,25,26,27,28,29]. Yet, the clinical utility of these markers remains to be determined.

In search of original biomarkers for PDAC, we started from the consideration that cell transformation by the *KRAS* oncogene generates protein dysregulation in the affected cells. It is well established that proteins secreted by cancer cells display several differences compared to normal counterparts, and secreted proteins may contribute to cancer-associated features [22,24,26,30].

*KRAS* mutations are the first genetic changes during disease progression, being commonly detected in early PanIN lesions, and are found in more than 95% of PDAC cases [31]. Constitutive activation of *KRAS* and persistent stimulation of downstream pathways sustains tumor cell proliferation, migration, metastasis and metabolic reprogramming, along with the evasion of the anti-tumor immune response [31,32,33,34,35,36,37,38]. With the idea that secreted proteins may reach the systemic circulation, and thus be detected in plasma, we tested whether candidate PDAC biomarkers can be identified in the secretome of recently transformed pancreatic cells transduced with the oncogenic *KRAS^G12V^.* We developed an in vitro model for KRAS transformation using the near-normal human pancreatic ductal epithelial cell line HPDE immortalized with the *HPV16-E6E7* genes (HPDE-E6E7) [39,40]. We focused our attention on the proteins secreted by HPDE-E6E7 cells transduced with a vector carrying the oncogenic *KRAS^G12^*^V^.

In this study, we have combined diverse experimental approaches: (i) proteomic analysis of the conditioned media of *KRAS^G12V^*-transduced pancreatic cell lines; (ii) transcriptomic analysis of PDAC tissues and the adjacent unaffected pancreas, and (iii) quantification of the circulating levels of selected mediators in the blood of PDAC patients and in patients with non-tumoral pancreatic pathologies. We validated a set of proteins associated with *KRAS^G12V^* expression in samples of PDAC patients, and further quantified some of these proteins in the circulating plasma of patients to verify their diagnostic potential.

## 2. Materials and Methods

### 2.1. Cell Culture and Lentiviral Transduction

HPDE cells were transduced with the *KRAS^G12V^* oncogene using a pRRLsinPPTGFPpre lentiviral vector, as previously described [41]. GFP-positive transduced cells were sorted by FACSAria, and immediately cloned by limiting dilution and expanded upon propagation. Cloned cell lines (E21, E30, E38, M7, M19 and M36) were cultured in RPMI 1640 medium supplemented with 10% FBS, 2 mM UltraGlutamine and 100 U/mL penicillin/streptomycin (Lonza, BioWhittaker, Rome, Italy). Phenotypic and functional characterization of the cloned KRAS-HPDE cell lines has been reported [41].

### 2.2. Proteomics Analysis by SILAC Mass Spectrometry

Stable Isotope Labeling by Amino Acids in Cell Culture (SILAC) and mass spectrometry methods are described in the detailed Appendix A.

### 2.3. RT-PCR and Quantitative Real-Time PCR

Total RNA was isolated from cells cultured in vitro and from surgical specimens of pancreatic cancer patients. RNA was isolated using the Promega SV Total RNA isolation system (Catalog number: Z3100) according to the manufacturer’s instructions. One microgram of total RNA was reverse transcribed using a High-Capacity cDNA Archive kit (Applied Biosystems, Monza, Italy) according to the manufacturer’s instructions. Tumor sample cDNA was analyzed using SYBR Green Quantitative Real-Time PCR on an ABI Prism^®^ 7900HT Fast Real-Time PCR System (Applied Biosystems). GAPDH was used as an internal control to normalize samples. All gene-specific primers were in-house designed and checked using Ab Applied Biosystems software. The sequences of the primer pairs were as follows: human GAPDH (sense: 5′-AGA TCA TCA GCA ATG CCT CCT G-3′; antisense: 5′-ATG GCA TGG ACT GTG GTC ATG-3′), human Laminin-C2 (sense: 5′-GTA TGT GAA CCC ACA ACC CAC AA-3′; antisense: 5′-TGT CCA CTG GCT TCT CAG GGT-3′), human Tenascin-C (sense: 5′-TCT CTG CAC ATA GTG AAA AAC AAT ACC-3′; antisense: 5′-TCA AGG CAG TGG TGT CTG TGA-3′), human RAN (sense: 5′-GGC GCT TCT GGA AGG AAC-3′; antisense: 5′-ACG TTT CAC GAA GGT CGT TT-3′), human Stanniocalcin 2 (sense: 5′-TCA AAG ACG CCT TGA AAT GTA A-3′; antisense: 5′-CAG TTC TGC TCA CAC TGA ACC T-3′), human Farnesyl pyrophosphate synthase (FPPS/FDPS) (sense: 5′-TCT CCC AGA TCG TTA GGG TG-3′; antisense: 5′-TCC CGG AAT GCT ACT ACC AC-3′), human Ubiquitin carboxyl-terminal hydrolase-L1 (UCHL-1) (sense: 5′-CCG AGA TGC TGA ACA AAG TGC-3′; antisense: 5′-CAT GGT TCA CCG GAA AAG GC-3′), human PTX3 (sense: 5′-CGA AAT AGA CAA TGG ACT CCA TCC-3′; antisense: 5′-CAG GCG CAC GGC GT-3′). The threshold cycle, Ct, was automatically provided using the SDS2.2 software package (Applied Biosystems).

### 2.4. Patients

Two hundred patients with pathologically confirmed pancreatic adenocarcinoma (PDAC) eligible for surgery were prospectively enrolled, upon the signing of an informed consent document; this study was approved by the Ethical Committee of the Clinical and Research Institute, Humanitas (Rozzano, Italy). All patients were enrolled from a single institution during the periods May 2012–December 2014 (*n* = 78) and April 2016–December 2017 (*n* = 122) (Table 1). We also evaluated 88 patients with other benign pancreatic diseases undergoing surgery; these included: cystadenoma (*n* = 21), chronic pancreatitis (*n* = 15), Intraductal Papillary Mucinous Neoplasms (IPMN) (*n*= 52). An institutional database was created to identify patients and to register clinico-pathological characteristics and outcomes. Regarding the follow-up of PDAC patients after surgery, patients were seen on an outpatient basis every 4–6 months. Disease status was assessed by serial CT scans and other diagnostic testing, as needed. Causes of death were assessed by examining medical records, by interviewing family doctors, or through death certificates. Follow-up cutoff date was December 2019.

### 2.5. Sample Collection and Plasma Preparation

All samples were collected following the institutional Biobank Standard Operating Procedures. Surgical specimens were collected from the surgical theatre immediately after excision and transferred to the pathology department. Pathologists selected tumor specimens necessary for the diagnosis, and the left-over tumor tissue and the non-involved pancreatic tissue were immediately frozen in liquid nitrogen. For each tumoral specimen, a specular sample was formalin fixed and paraffin embedded for quality control. Snap-frozen samples were transferred and stored at −80 °C in 2 mL barcoded cryovials. Tissue samples were frozen within 30 min from excision. Peripheral blood samples (15 mL) from patients were collected into BD Vacutainer EDTA tubes and centrifuged at 1700 rpm at room temperature for 15 min. Plasma was aliquoted into 0.5 mL barcoded cryovials and stored at −80 °C. Blood samples were kept at 4 °C and processed within 2 h from collection. Sample data, including collection and preservation time, were recorded in the Biobank Management System.

### 2.6. Immunohistochemistry

Immunohistochemistry was used to study the expression of LAMC2, TNC and PTX3 in paraffin-embedded PDAC patient tissues. Harvested PDAC tumor tissues were fixed in 4% paraformaldehyde (PFA) at 4 °C overnight and embedded in paraffin. Sections (2 μm) were deparaffinized and incubated in a continuous gradient of alcohol and soaked in 10 mM citrate buffer (pH 6) in a microwave oven at 600 W thrice for 5 min for antigen retrieval. For PTX3, sections were soaked in 0.25 mM EDTA buffer (pH 8) at 98 °C in a water bath for 20 min. The slides were then blocked with peroxidase I (Biocare Medical, Pacheco, CA, USA) for 30 min at room temperature (RT). Non-specific binding sites of each section were blocked with 2% BSA + 0.02% NP40 in PBS for 30 min at RT, followed by incubation at 4 °C overnight with primary antibody anti-human TNC (ab6393, Abcam, Milan, Italy), anti-human LAMC2 (17370002, Novus Biologicals, Milan, Italy) and anti-human PTX3 (affinity-purified rabbit IgG anti-human PTX3 developed in-house, final concentration 2 μg mL^−1^) [42] in PBS, followed by incubation with secondary antibody horseradish peroxidase-labeled anti mouse/rabbit IgG (Biocare Medical) for 30 min at RT. Finally, sections were visualized under a bright-field microscope with a Betazoid DAB Chromogen kit (Biocare Medical), counterstained with hematoxylin, and mounted using EuKitt^TM^ quick-hardening mounting medium (Sigma-Aldrich, Milan, Italy). Sections were analyzed using a VS-120 dotSlide system (Olympus, Milan, Italy). Immunohistaining was conducted on the area with the most representative histological lesion. Staining distribution was analyzed both as cellular positivity and extracellular stromal expression. The adjacent non-affected pancreatic parenchyma was used as a control for normal expression.

### 2.7. ELISA

Quantitative ELISAs were performed on freshly defrosted plasma samples and tested in triplicate. For LAMC2 and TNC, commercial ELISA kits were used following manufacturer’s instructions: hLaminin-C2, ELISA Kit Code n. LS-F33142; hTenascin-C, ELISA Kit Code n. LS-F27283 (LSBio, Seattle, WA, USA). For PTX3, a sandwich ELISA (detection limit 100 pg/mL, inter-assay variability 8–10%), developed in-house, was used, as previously described [43,44]. No cross-reaction with human CRP was observed. Plasma levels of the candidate biomarkers were quantified in patients with pathologically confirmed PDAC and in patients with benign pancreatic pathologies, as specified. Blood samples from healthy individuals (*n* = 135, age range 24–65 years) were collected from blood donors and laboratory members.

### 2.8. Statistical Analysis

Prism software (v6.0 a; GraphPad) was used to conduct appropriate statistical procedures, as specified in figure legends. Statistical analysis of ELISA values among different groups was performed via one-way ANOVA and parametric t-tests with Welch’s correction; outliers were removed using the ROUT method. A *p* value < 0.05 was considered significant unless noted otherwise. Overall survival time was calculated from the date of surgery to the date of death or last contact. Survival curves were calculated using the Kaplan–Meier method and the log rank test to compare curves between the two different groups.

## 3. Results

### 3.1. Identification of Secretome Perturbations Induced by the Oncogenic KRAS^G12V^ Mutation in Human Pancreatic Cells

With the aim of investigating the effects of the earliest oncogenic mutation on the secretome of pancreatic cells, we set up a model of normal human pancreatic ductal cells growing in vitro to be infected with the cDNA of the oncogenic *KRAS^G12V^* cloned into a GFP lentiviral vector (pRRL.sinPPT.CMV.GFPpre-K-RAS^G12V^). The first attempt was to use primary cultures of ductal epithelial cells from normal human pancreatic samples (adjacent to pathological pancreatic tissues undergoing surgery). However, this approach failed; in two separate experiments, the proportion of GFP+ KRAS^G12V^-transduced cells was only around 26% (Appendix A). Furthermore, cells had a very limited life span and stopped proliferating in a few days, likely due to the mechanism of oncogene-induced senescence [45], and eventually died. We therefore turned to a well-known cellular model, an established normal human pancreatic cell line of ductal epithelial origin (HPDE), which had been immortalized with E6/E7genes of the papilloma virus-16 [39,40]. HPDE cells were first transduced with the KRAS^G12V^ vector, and then selected by flow cytometry for GFP+ cells (Appendix A). Mock infection of HPDE cells was also performed to be used as a control. Both Mock and KRAS-HPDE cells regularly grew in vitro, but only the KRAS-mutated cells were able to generate tumors in vivo in SCID mice (Appendix A).

Next, to gain insight into the impact of KRAS-induced phenotype modifications, we immediately cloned the KRAS-HPDE cell line and selected different cloned lines for phenotype and functional characterization [41]. In the 80 cloned cell lines obtained, we observed a heterogeneous degree of epithelial-to-mesenchymal transition, with about half of the clones revealing a mesenchymal phenotype, 25% an epithelial phenotype, and the rest a mixed phenotype (not shown). Notably, all the selected cloned cell lines were able to generate tumors in mice, but only those with a mesenchymal phenotype metastasized to the liver in a spleen–liver mouse model [41]. We therefore selected—as the source of potential biomarkers hopefully encompassing the heterogenous effects of the *KRAS* oncogene—well-characterized cloned cell lines with different phenotypes (i.e., three mesenchymal and three epithelial cell lines).

To analyze the qualitative/quantitative secretome perturbations induced by oncogenic *KRAS*, we used a global quantitative proteomic approach based on SILAC MS (Appendix A). Briefly, the secretome of each KRAS-cloned cell line was co-analyzed for direct comparison, throughout all analytical steps, with the Mock secretome, to obtain, for each identified protein, a log2 ratio (Clone-H/Mock-L), where H and L refer to the different stable isotope labeling of the samples. Our proteomic analysis overall identified/quantified a total of 600 non-redundant proteins across the nine analyzed secretomes (three Mock-H/Mock-L and six KRAS-clones-H/Mock-L sample pairs) (complete dataset in Appendix A). Across the six KRAS-cloned cell lines, the proteomic analysis identified 163 dysregulated proteins (Appendix A). For half of these proteins, dysregulation occurred in one cell line only. Of the remaining 83 proteins with multiple dysregulations across the clones, 1 protein (Clusterin) was dysregulated in all clones, 12 proteins in four or five clones, and 70 in two or three clones. For the large majority (80%) of the proteins, the observed dysregulations across the clones were convergent. Figure 1A,B shows the multiple convergent dysregulations observed (29 all up- and 38 all downregulated proteins, respectively), while Appendix A shows the 16 proteins with divergent dysregulations [46]. 

A heatmap of all the 163 dysregulated proteins—hierarchically clustered according to their log2 ratio (H/L) values across the six KRAS-clones—is presented in Appendix A. Furthermore, an overall view of all secretome profiles can be found in the Appendix A, where different heatmaps examine the different zones of the secretomes (stable or perturbed) in all the six KRAS-clones vs. the three Mock samples.

Among the 29 proteins upregulated in at least two KRAS-clones, we decided to eliminate proteins that, according to literature data, are known to be widely expressed in several tissues/tumors, and are this likely to be non-specific for PDAC, such as: the metalloprotease MMP9, the carboxypeptidase CPVL and prostaglandin E synthase (PTGES3). We focused our attention on six proteins: two extracellular matrix (ECM) proteins, i.e., Laminin-C2 (LAMC2) and Tenascin-C (TNC); two proteins related to KRAS activity, i.e., RAS-related Nuclear protein (RAN) and Farnesyl pyrophosphate synthase (FPPS/FDPS); Stanniocalcin2 (STC2), a multifunctional glycoprotein with recently recognized role in cancer; Ubiquitin carboxy-terminal hydrolase L1 (UCHL-1), a deubiquitinating enzyme that was the most consistently upregulated protein among our six KRAS-cloned lines. Pentraxin 3 (PTX3), a member of the pentraxin superfamily induced in response to primary inflammatory signals [39,40], was finally added to our panel of candidate biomarkers. We had in fact previously found, via specific ELISAs, that PTX3 was secreted in large amounts by four of the KRAS-clones, but not by Mock cells [41]. For the seven selected candidate biomarker proteins, we confirmed that mRNA expression levels in the six KRAS-cloned lines were higher compared to Mock cells (Appendix A).

### 3.2. Expression of the Cancer-Associated Proteins in Pancreatic Cancer tissues

To verify the expression of our seven-protein panel in pancreatic cancer tissues, we studied frozen pancreatic tumor specimens (*n* = 20 cases) and non-involved pancreatic tissues (*n* = 16) collected during surgery and stored in our institutional Biobank. Of the seven selected genes, LAMC2, FPPS, TNC, RAN and STC2 had significantly higher mRNA levels in PDAC samples compared to the normal adjacent pancreatic tissues, and PTX3 showed a trend approaching significance (*p* = 0.08), while RNA levels of UCHL-1 were not significantly increased in tumor tissues. (Figure 2A).

On the basis of these findings, as well on the availability of ELISAs for these proteins, we selected three candidate biomarkers, LAMC2, TNC and PTX3, for further study. We first verified protein expression in pancreatic tumor tissues via immunohistochemistry. Four different tumor samples were immunostained, and the representative images show that LAMC2 was clearly detected in tumor cells, and was particularly intense in some samples (Figure 2B and Appendix A); the expression pattern of TNC was heterogenous, as in some samples the protein was expressed both in neoplastic cells and in the stroma, while in other samples, cancer cells were negative whereas a strong signal was present in stromal fibers; immunostaining of PTX3 was detected in some cancer cells, and occasionally in stromal cells. Notably, all markers were negative in the adjacent non-involved pancreas (Figure 2B and Appendix A).

### 3.3. Plasma Levels of Candidate Biomarkers in PDAC Patients

Our major goal was to quantify the circulating levels of the three proteins in pancreatic cancer patients to verify whether their relative abundance could be exploited as a biomarker of disease in a non-invasive blood test. Blood samples were collected on the day of surgery, processed within 2 h, and stored at −80 °C, according to the Biobank Standard Operating Procedures. Plasma samples from 200 prospectively enrolled PDAC patients were collected, upon informed consent, at a single institution in two different periods: from May 2012 to December 2014 (*n* = 78 patients) and from March 2016 to December 2017 (*n* = 122 patients). The samples were analyzed for the quantification of biomarker proteins at different time points, and used to generate two distinct cohorts of patients: a training set and a validation set. The clinico-pathological characteristics of PDAC patients are listed in Table 1. Blood samples from healthy individuals (*n* = 135, age range 24–65 years) were tested in parallel in the training set (*n* = 53) and validation set (*n* = 82).

The ELISAs revealed that the plasma levels of LAMC2 were significantly higher in pancreatic cancer patients compared to healthy donors, both in the training set and in the validation set (*p* < 0.0001) (Figure 3A). Considering all patients, the median LAMC2 concentration was 2.53 ng/mL in PDAC patients (range 0.1–62.5 ng/mL), while in healthy individuals, it was <0.001 ng/mL (range 0.001–19.2 ng/mL).

Levels of PTX3 were significantly higher in PDAC patients in the training and validation set (*p* < 0.0001) (Figure 3B). Median PTX3 concentration was 7.61 ng/mL (range 2.35–70.03 ng/mL) in patients, and 1.65 ng/mL (range 0.43–4.62 ng/mL) in healthy donors. Furthermore, for TNC, the plasma values were significantly higher in cancer patients (*p* < 0.0001) (Figure 3C). Median TNC concentration was 8.35 ng/mL (range 0.5–49.37 ng/mL) in patients, and 4.49 ng/mL (range 0.01–13.05 ng/mL) in healthy donors.

Sensitivity and specificity of the candidate biomarkers LAMC2, PTX3 and TNC were analyzed using the Area Under the Curve (AUC) of the Receiver Operating Characteristics (ROC) curve. We assessed their individual performance by discriminating between PDAC patients and healthy individuals. LAMC2 was able to distinguish between the two groups, with an AUC of 0.84 (95% CI: 0.78–0.90). With a cutoff value of 0.1 ng/mL, specificity was 0.63 and sensitivity was 1 (Figure 4A). We next examined the association of LAMC2 with patient survival rate. Of the 200 PDAC patients examined, 148 underwent surgery, while 52 patients were considered non-resectable at the time of surgery because of locally advanced or metastatic disease, and were not considered for the survival analysis. In the 148 resected patients, the median overall survival was 27.5 months. PDAC patients with low circulating levels of LAMC2 (<1.37 ng/mL) had a significantly longer median survival time of 51.4 months, compared to patients with LAMC2 >1.37 ng/mL, which had a median survival time of 25.9 months (*p* = 0.0007) (Figure 4B).

For PTX3, the AUC was 0.99 (95% CI: 0.98–0.99). With a cutoff value of 2.35 ng/mL, specificity was 0.85 and sensitivity was 1 (Figure 4C). With a threshold value of 2.35 ng/mL, the survival curves of patients with high or low PTX3 levels were not significantly different (*p* = 0.84). Even after increasing the concentration value of PTX3 to 6.0 ng/mL, the survival curves were not statistically different (*p* = 0.65) (Figure 4D). TNC had the worst performance, with a specificity of 0.67 (95% CI: 0.61–0.73) and poor sensitivity (Appendix A), and was not associated with patient survival. We conclude that PTX3 exhibits good performance as a biomarker of disease, with high specificity and sensitivity, but is not able to predict patient survival. Instead, LAMC2 has moderate specificity and sensitivity, but its high circulating levels are strongly associated with a shorter survival time.

### 3.4. Biomarker Panel in Early-Stage PDAC Patients and Patients with Low Levels of CA19-9

We next considered whether the identified biomarkers could stratify PDAC patients according to the stage of the disease. This is of particular importance in pancreatic cancer, as surgery requires the detection of a small tumor not yet progressed. Patients were divided into early-stage tumors (I/IIA and IIB) and late-stage tumors (III and IV), as detailed in Table I. Levels of LAMC2 were significantly higher in early-stage I/IIA and IIB PDAC (*n* = 127) than in healthy individuals (**** *p* < 0.0001) (Figure 5A). Similarly, in early-stage tumor patients, PTX3 levels were higher than controls (**** *p* < 0.0001) (Figure 5B), indicating that both LAMC2 and PTX3 are already altered in the earliest stages of PDAC.

CA19-9 is the only serological tumor biomarker approved by the FDA for pancreatic cancer, which is far from ideal. CA19-9 may give false positive results in benign pancreaticobiliary diseases, and is not expressed in about 10% of the Caucasian population [11,12,13,14,15]. Considering the levels of CA19-9 in our cohort of 200 patients, 58 cases (29%) had CA19-9 levels below the standard cutoff value of 37 U/mL. We then analyzed the levels of LAMC2 and PTX3 in CA19-9-low patients. All CA19-9-low patients had LAMC2 and PTX3 levels significantly higher than healthy donors (*p* < 0.0001) (Figure 5C,D). These results indicate that these two candidate biomarkers show high performance in detecting PDAC patients, even in those cases where the standard biomarker CA19-9 was below the cutoff value.

### 3.5. Biomarker Panel Expression in Plasma of Patients with Other Pancreatic Pathologies

We next explored the quantification of biomarkers in plasma samples of patients with benign pancreatic tumors (cystadenoma, *n* = 21), in patients with chronic pancreatitis undergoing surgery (*n* = 15), and in patients diagnosed with Intraductal Papillary Mucinous Neoplasm (IPMN) (*n* = 52), a recognized pre-cancerous condition that is important to detect [47,48]. Levels of LAMC2 were significantly higher in cystadenoma and pancreatitis patients compared to healthy donors (*p* < 0.0001) (Figure 6A). PTX3 was significantly higher in pancreatitis (*p* < 0.05), but not in cystadenoma (Figure 6B). Patients with IPMN had very high levels of PTX3 compared to healthy donors (*p* < 0.0001) (Figure 6B), but not of LAMC2 (Figure 6A). These findings suggest that a pathological derangement in pancreatic tissues already triggers the release of these two proteins into the bloodstream (in particular, LAMC2 in cystadenoma and pancreatitis, and PTX3 in IPMN), and that further diagnostic imaging is warranted. It must be noted, however, that the biomarker levels were significantly more elevated in patients with PDAC compared to patients with non-neoplastic conditions, as depicted in Figure 6C,D, with two exceptions: pancreatitis patients had high levels of LAMC2 that were similar to levels in PDAC patients, and PTX3 was already high in IPMN patients. This latter finding suggests that PTX3 quantification cannot discriminate between patients with IPMN or PDAC, but could be useful as a potential biomarker of pancreatitis.

## 4. Discussion

In this study, we have defined a proteomic signature of secreted proteins induced by the oncogenic *KRAS^G12V^* in human pancreatic cells. It is well established that proteins secreted by tumor cells may contribute to cancer-associated features, and may be a source of candidate biomarkers of disease [22,26]. The secretome analysis was performed on six different subcloned lines from the KRAS-transduced pancreatic HPDE cells, displaying specific features of epithelial or mesenchymal phenotypes, as depicted in Appendix A, and as previously reported [41]. The numerous secretion alterations induced by oncogenic *KRAS* were heterogeneously represented across the six studied secretomes, with no evident relation with their epithelial vs. mesenchymal phenotype. It is likely that other sets of non-secreted proteins, or proteins escaping detection in this study, may be dysregulated by KRAS in accordance with the different characteristics of our subcloned lines [41].

Among the upregulated proteins, we found proteins related to the RAS pathway, such as RAN and FPPS/FDPS, and proteins related to the ECM or the cell–matrix interface, such as LAMC2 and TNC. RAN is a RAS-related GTPase with numerous cellular functions, including the transportation of molecules between the nucleus and cytoplasm, the regulation of cell cycle-related proteins, and the assembly of the mitotic spindle. Ran GTPase is overexpressed in several cancer types, and its high levels correlate with malignant features [49,50,51]. Another molecule at the crossroads of the KRAS pathway is the farnesyl pyrophosphate synthase (FPPS). FFPS is a crucial enzyme of the complex mevalonate biochemical network that leads to cholesterol biosynthesis and several other metabolites, which play an essential role in cell viability, signaling and proliferation. Interestingly, FPPS is involved in the prenylation/farnesylation of small GTPase, such as RAS. As the mevalonate pathway is often dysregulated in cancer cells, FPPS levels were found to be elevated in several tumors. FPPS has not been studied as biomarker of PDAC, but this molecule has been the object of potential therapeutic interest because a number of available drugs target the mevalonate pathway, including the bisphosphonates (directly inhibiting FPPS) and the statins (inhibiting the HMG-CoA reductase) [52,53]. In fact, the anti-neoplastic effects observed with bisphosphonates in some studies have been attributed to multiple mechanisms related to the FPPS enzyme, which may include the downregulation of prenylated-KRAS with cell growth inhibition effects [54]. Furthermore, statins have been demonstrated to inhibit RAS prenylation and downstream signaling pathways, including ERK1/2, Akt and mTOR, therefore impacting cell survival and proliferation [55]. Here, we found that these two RAS-related proteins were elevated in PDAC tissues, compared to adjacent non-involved tissues.

Two other secreted proteins induced by the *KRAS* cellular transformation, TNC and LAMC2, were identified in this study. TNC is a large ECM protein that interacts with many cell surface receptors, including integrins and Annexin A2; it modulates cell signaling and influences cell migration and proliferation [56]. TNC is mainly produced by pancreatic stellate cells in the stroma; notably, we detected TNC in *KRAS*-transformed epithelial pancreatic cells cultured in vitro, in the absence of mesenchymal cells. TNC expression has been detected in pancreatic tissues and is upregulated in the progression from pancreatic intraepithelial neoplasia (PanIN) to PDAC [56,57]. TNC, a major component of the cancer-specific matrix, has been found overexpressed in solid tumors, and its expression is associated with poor prognosis in several cancers, including PDAC [58,59]. The effect of TNC on cancer cells encompasses its ability to activate the oncogenic Wnt/β-catenin and YAP/TAZ signaling pathways [60,61]. Balasenthil et al. reported that plasma levels of TNC measured by ELISA were higher in pancreatic cancer patients than healthy controls, and that TNC improved the performance of the biomarker CA19-9 in all early-stage patient cohorts [62]. In our study, we found significantly increased expression of TNC in PDAC tissues compared to normal tissues, and in patients’ plasma relative to healthy controls. The ROC curve, however, lacked high specificity and sensitivity, and the plasma levels of TNC were not associated with patient survival.

The other matrix-related protein, LAMC2, appeared more promising as a biomarker in PDAC. Laminins are a family of ECM glycoproteins representing the major non-collagenous component of the anchoring filaments that connect epithelial cells to the underlying basement membrane, and are crucially involved in tissue organization, including the specification of epithelial features of cell polarity [63]. Laminins have been implicated in many tumor-related processes, such as cell adhesion, migration, differentiation and metastasis. LAMC2 is the gamma 2 chain of Laminin-332, formerly known as Laminin-5. LAMC2 and other family members have been found overexpressed in various cancer types [64,65,66]. Gene silencing of the *LAMC2* gene in pancreatic cancer cell lines inhibited proliferation, migration and invasion ability in vitro and enhanced sensitivity to gemcitabine [67]. These and other findings reveal its crucial role in regulating cell functions at the epithelial–stromal interface, and suggest that LAMC2 may be considered as a therapeutic target [67,68]. A previous study identified LAMC2 in a proteomic analysis of PDAC cancer tissues [69], and high circulating levels of LAMC2 have been described [70]. LAMC2 overexpression in PDAC tissues has been proposed as an indicator of poor prognosis [71,72]. In our study, we confirm that LAMC2 is overexpressed in PDAC tissues compared to the adjacent pancreas, and that its circulating levels are significantly higher in PDAC patients relative to healthy controls. The ROC curve for LAMC2 showed good specificity and sensitivity as a biomarker of disease and, most importantly, high levels of LAMC2 identified the group of patients with the worst survival. Therefore, our results with a new cohort of prospectively collected PDAC patients support the finding that LAMC2 is a potential diagnostic biomarker of this disease.

The molecule PTX3, originally cloned by our group in the early 1990s [73], was also represented in our secretome analysis via ELISA. Pentraxins are a family of evolutionarily conserved molecules with diverse roles in innate immunity and inflammation. C-reactive protein (CRP) and serum amyloid P component (SAP) are the short, or “classical”, pentraxins, which are mainly produced by hepatocytes as acute phase proteins [74]. PTX3 is the prototype of the long pentraxin subfamily, originally identified as an IL-1 or TNF-inducible gene, and is produced by different cell types in response to pro-inflammatory stimuli and microbial moieties [74,75]. In addition to providing defense against infectious agents, PTX3 plays several functions in tissue repair and in the regulation of cancer-related inflammation [75]. PTX3 is expressed in inflammatory conditions, and acts as a tuner of complement-activation and leukocyte recruitment. For example, by interacting with other pattern recognition molecules, PTX3 amplifies the activation of innate immune responses, whereas by interacting with fibrinogen/fibrin and collagen, PTX3 modulates injury-induced responses and favors fibrinolysis, contributing to tissue remodeling and repair [74,75]. Furthermore, PTX3 binds selected fibroblast growth factors (FGFs), including FGF2 and FGF8b, and inhibits FGF-dependent angiogenic responses [76]. Interestingly, tissue remodeling and vascular inhibition are two conditions that bear relevance in the strong desmoplastic reaction associated with pancreatic cancer [77].

In PDAC tissues, PTX3 was previously found expressed by mesenchymal stellate cells in the stroma [78]; however, in our study, PTX3 was overexpressed in cultured *KRAS*-transduced epithelial pancreatic cells, demonstrating that cancer cells may also produce PTX3 upon *KRAS* transformation. The mRNA levels of PTX3 were higher in PDAC tissues, although only approaching statistical significance (*p* = 0.08). Due to the inflammatory nature of this protein [74,75], we suspect that the adjacent pancreatic tissue, used as a reference, was already involved in a process of cancer-associated inflammation. This was also the case for other inflammatory *KRAS*-induced cytokines, such as IL-6 and CXCL8 (not shown). The plasma levels of PTX3 were significantly higher in PDAC patients relative to controls, and the ROC curve showed very high sensitivity and specificity. In line with these findings, in a recent collaborative paper, Goulart et al. reported that high levels of serum PTX3 had a better predictive value for the detection of PDAC than serum CA19-9 and CEA [78]. Unlike what we observed with LAMC2, we found that circulating levels of PTX3 were not different in patients with short or long overall survival. In contrast, Kondo et al. reported that, in a cohort of 78 advanced PDAC patients, high PTX3 levels identified the patients with shorter overall survival [79]. Of note, PTX3 has also been investigated as a biomarker of acute pancreatitis, with higher plasma concentrations in moderate and severe pancreatitis compared to mild disease, but with a poor performance—inferior to CRP—in predicting which patients would progress to a systemic inflammatory syndrome with fatal outcomes [80]. Overall, we conclude that PTX3 is expressed in PDAC tissues, and is produced by stromal and cancer cells, likely in response to an associated inflammation that characterizes *KRAS*-harboring tumors, or in response to local tissue damage.

Considering our best candidate biomarkers, LAMC2 and PTX3, it is interesting to note that both were already significantly elevated in patients with early-stage PDAC, underscoring their potential to detect patients eligible for surgical resection, which is currently the best potentially curative medical intervention available for these patients [4,6,7,8,9]. In addition, patients with CA19-9 below the standard cutoff value of 37 U/mL had LAMC2 and PTX3 levels above the median values of healthy donors. This result indicates that those patients with a “negative” score for the routinely used diagnostic marker (CA19-9) may be detected by these candidate biomarkers and, therefore, LAMC2 and PTX3 can improve the sensitivity of CA19-9 for the clinical diagnosis. Finally, in this study, we also considered patients with benign pancreatic pathologies, such as cystadenoma of the pancreas, chronic pancreatitis and IPMN, that were elected for surgery. The circulating levels of LAMC2 were significantly higher in patients with cystadenoma and chronic pancreatitis than in healthy donors, and PTX3—but not LAMC2—was detected in patients with IPMN. Given that false positive values represent a major concern for a diagnostic biomarker, and that plasma levels of LAMC2 in patients with cystadenoma—as well as PTX3 levels in patients with chronic pancreatitis—were still significantly lower than in PDAC patients, we propose that intermediate levels of LAMC2 and PTX3 (higher than in healthy donors, but lower than in PDAC patients) may identify patients for which further diagnostic imaging is warranted.

## 5. Conclusions

In this study, we show that LAMC2 and PTX3 are overexpressed in cultured pancreatic cells after transduction with oncogenic *KRAS^G12V^*, and in surgical tissues of patients with PDAC. These two proteins can be detected via ELISA in the systemic circulation of patients, and their levels are significantly higher than in healthy individuals. In particular, both LAMC2 and PTX3 are already differentially expressed in patients with early-stage PDAC, and in patients with CA19-9 levels below the standard threshold. In conclusion, circulating levels of LAMC2 and PTX3 may increase the predictive power of CA19-9, and may have potential clinical utility for the early diagnosis of surgically resectable PDAC.

## Figures and Tables

**Figure 1 cancers-14-02653-f001:**
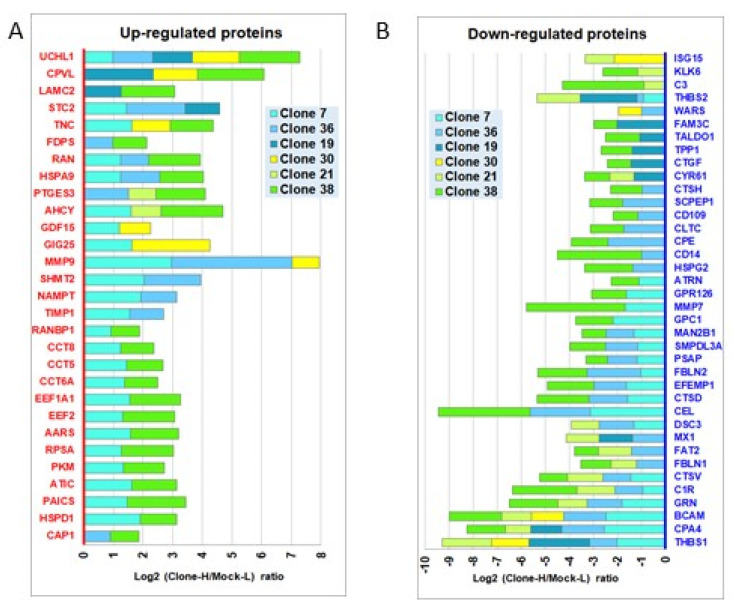
Distribution of multiple converging up- or down-dysregulations of secreted proteins across the 6 KRAS-clones (H), each paired with a Mock (L) sample, obtained by SILAC MS quantitative proteomics. Cutoff for dysregulation: log2 ratios (H/L) exceeding ±0.8916. (**A**) All the 29 upregulated proteins. (**B**) All the 38 downregulated proteins.

**Figure 2 cancers-14-02653-f002:**
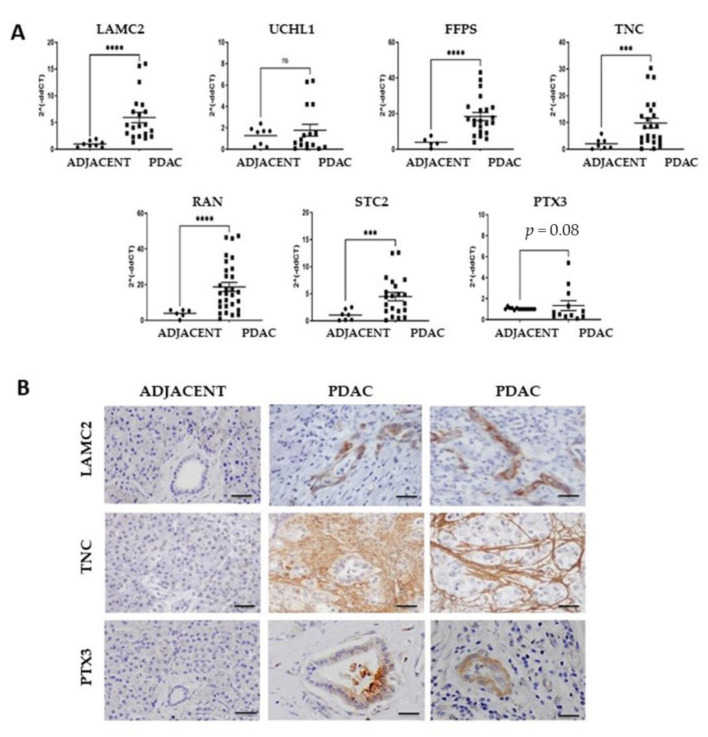
Expression of candidate biomarker proteins in surgical PDAC tumor samples. (**A**) mRNA expression levels of LAMC2, TNC, PTX3, FPPS, UCHL-1, STC2 and RAN in surgical specimens of PDAC (*n* = 21) and in the non-involved adjacent pancreas (*n* = 16). Statistical analysis: parametric t-test with Welch’s correction, *** *p* < 0.01, **** *p* < 0.001. (**B**) Representative images of immunohistochemistry for LAMC2, TNC and PTX3 in PDAC surgical samples and adjacent pancreatic tissues showing positive immunostaining in tumor cells and also for TNC in the stroma.

**Figure 3 cancers-14-02653-f003:**
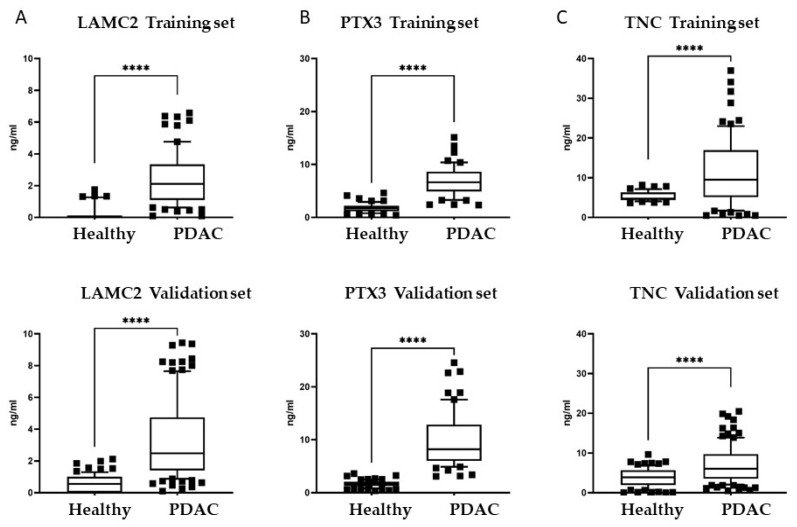
Plasma levels of LAMC2, PTX3 and TNC in pancreatic cancer patients and in healthy donors. Two cohorts were tested, Training set: PDAC *n* = 78 patients, Healthy *n* = 53 donors; and Validation set: PDAC *n* = 122 patients; Healthy *n* = 82 donors. Plasma concentrations were measured by ELISA: (**A**) LAMC2, (**B**) PTX3 and (**C**) TNC. Statistical analysis: parametric t-test with Welch’s correction, outliers eliminated with «Identifying outliers» ROUT, **** *p* < 0.0001. Boxes correspond to values comprised between 10 and 90 percentiles.

**Figure 4 cancers-14-02653-f004:**
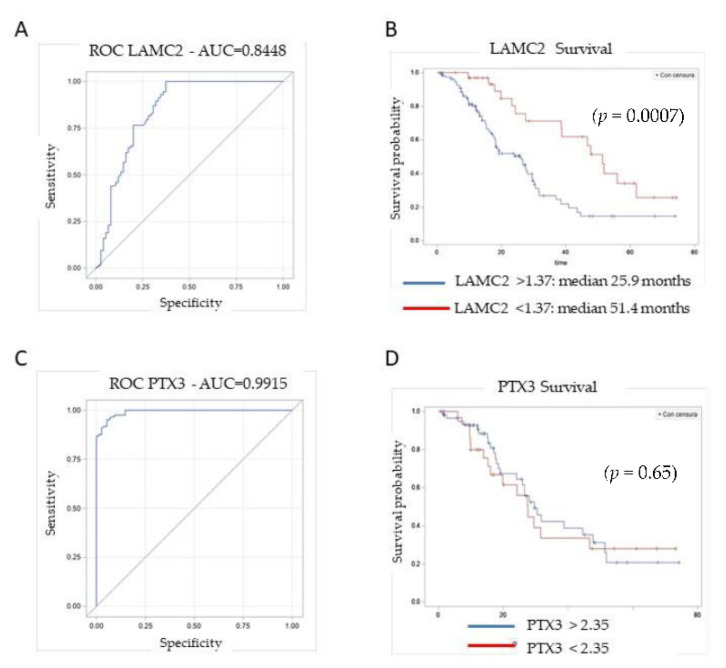
Sensitivity and specificity of (**A**) LAMC2 and (**B**) PTX3, analyzed by the Area Under the Curve (AUC) of the Receiver Operating Characteristics (ROC) curves. Analyses include PDAC patients (*n* = 200) and healthy individuals (*n* = 75). Association of biomarker levels of (**C**) LAMC2 and (**D**) PTX3 with patient survival rate, considering 148 effectively resected PDAC patients (52 were considered non-resectable at the time of surgery). PDAC patients with circulating levels of LAMC2 <1.37 ng/mL had a significantly longer median survival time of 51.4 months compared to patients with LAMC2 >1.37 ng/mL, which had a median survival time of 25.9 months. Statistical analysis: Kaplan–Meier method (*p* = 0.0007). For PTX3, with a threshold value of 2.35 ng/mL, the survival curves of patients with high or low PTX3 levels were not significant (*p* = 0.84).

**Figure 5 cancers-14-02653-f005:**
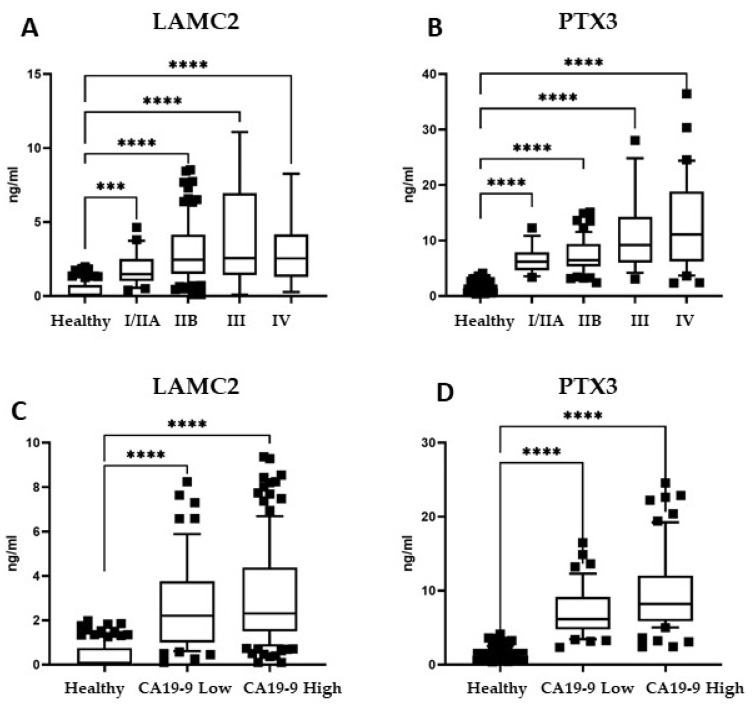
Biomarker panel quantification in early-stage PDAC patients and in patients with low levels of CA19.9. Plasma levels of (**A**) LAMC2 and (**B**) PTX3 quantified by ELISA in PDAC patients according to their stage, as detailed in Table I. Plasma levels of (**C**) LAMC2 and (**D**) PTX3 quantified by ELISA in PDAC patients with low CA19-9 values of <37 U/mL (*n* = 58 patients) and in PDAC patients with high CA19-9 values of >37 U/mL (*n* = 142 patients). Results are analyzed in comparison with the total number of healthy volunteers (*n* = 135). Statistical analysis: one-way ANOVA, outliers eliminated with «Identifying outliers» ROUT, *** *p* < 0.001, **** *p* < 0.0001. Boxes correspond to values comprised between 10 and 90 percentiles.

**Figure 6 cancers-14-02653-f006:**
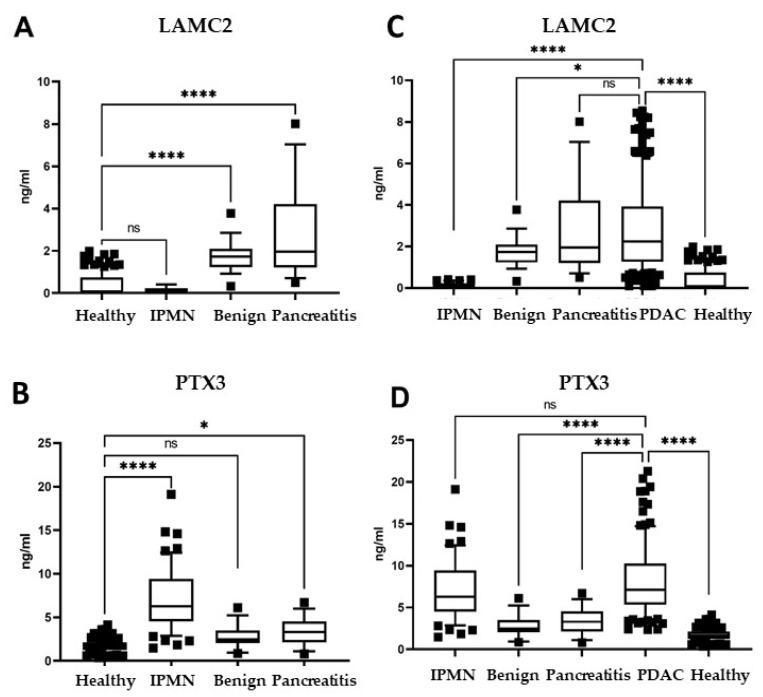
Biomarker panel expression in plasma of patients with non-malignant pancreatic pathologies. (**A**,**B**) Plasma levels of LAMC2 and PTX3 obtained by ELISA in patients with benign pancreatic tumors (cystadenoma, *n* = 21), chronic pancreatitis undergoing surgery (*n* = 15), or Intraductal Papillary Mucinous Neoplasms (IPMN, *n* = 52), relative to healthy individuals (*n* = 135). (**C**,**D**) Plasma levels of LAMC2 and PTX3 in non-malignant pancreatic patients relative to patients with PDAC (*n* = 200). Statistical analysis (**A**,**C**,**D**): one-way ANOVA; (**B**) Student’s t-test with Welch’s correction, outliers eliminated with «Identify outliers» ROUT, * *p* < 0.05, **** *p* < 0.0001. Boxes correspond to values comprised between 10 and 90 percentiles.

**Table 1 cancers-14-02653-t001:** Clinical characteristics of the study population of PDAC patients.

	Training Set	Validation Set
Recruiting period	2012–2014	2016–2017
PDAC patients *n*°	78	122
Gender	F (41); M (37)	F (69); M (53)
Age range	38–86	38–83
Stage		
I/IIA	15	14
IIB	53	45
III	0	19
IV	8	34
ND ^1^	2	10
Resected/Total	68/78	80/122

^1^ Not determined.

## Data Availability

The data presented in this study are available upon reasonable request.

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
