# Peer review of "Oncogenic KRAS-Induced Protein Signature in the Tumor Secretome Identifies Laminin-C2 and Pentraxin-3 as Useful Biomarkers for the Early Diagnosis of Pancreatic Cancer"

_cancers, 2022, doi:10.3390/cancers14112653_

Round 1

Reviewer 1 Report

This is a clearly described effort to identify additional serum markers predicative of early pancreatic cancer. Because of the need for such markers this is a highly relevant study. The methods are generally appropriate and the conclusions are justified by the data presented.

One weakness is limiting the analysis of proteins from the discovery stage to a select group (LAMC2, TNC and PTX3) for the patient serum studies when others appeared to equally or more significantly differentially expressed. Presumably this was due to absence of a readily available ELISA assay for the others. The authors should discuss this in more detail. The analysis of the value of the markers in serum with low CA19 is a strength of the study.

Author Response

Reply:

We thank the reviewer for her/his appreciation of our work. We have included in the manuscript a sentence commenting why we did not chose proteins that were highly upregulated. The sentence is at line  275-279.

“Among the 29 proteins upregulated in at least two KRAS-Clones, we decided to eliminate proteins that from literature data are known to be widely expressed in several tissue/tumors and likely are non-specific for PDAC, such as: the metalloprotease MMP9, the carboxypeptidase CPVL, the prostaglandin E synthase (PTGES3) and the chemokine CXCL3. We focused our attention on six proteins:…”

Reviewer 2 Report

This paper is a well-designed thoughtful approach to identify additional protein markers for detecting early pancreatic cancer (PDAC). The authors construct K-ras mutant cell-lines and determine protein levels compared with mock constructs.  They identify 7 proteins of interest, 3 of which are secreted, and 2 of these demonstrate potential utility to augment identification of early PDAC in CA19-9 low patients.  The experimental section is comprehensive, and the discussion is generous in citing earlier work, while not over-reaching in describing the importance of the current work.  This leads the reader to trust the validity of the work in a field rife with protein biomarkers that fail to live up to their original billing.  A very minor point, the most unregulated protein in Fig. 1 is MMP9, and perhaps the authors can add a sentence or two on why this was not pursued further (understood it is increased in many cancers, and thus non-specific).  Further, the authors may want to add the caveat that high levels of LAMC2 or PTX3 in CA19-9 low individuals may be due to other cancers or other causes of inflammation.  Overall, a first-rate paper.

Author Response

Reply:

Thank you for your appreciation. At lines 275-279 we have included in the manuscript a sentence commenting why we did not chose proteins that were highly upregulated.  “Among the 29 proteins upregulated in at least two KRAS-Clones, we decided to eliminate proteins that from literature data are known to be widely expressed in several tissue/tumors and likely are non-specific for PDAC, such as: the metalloprotease MMP9, the carboxypeptidase CPVL, the prostaglandin E synthase (PTGES3) and the chemokine CXCL3. We focused our attention on six proteins:…”

Regarding the point of high levels of LAMC2 or PTX3 in CA19-9 low individuals, just a specification: these patients were in fact diagnosed of PDAC and underwent surgical resection. Early stage PDAC is not uncommon to have low levels of CA19-9. The role of inflammation as causative effect of high PTX3  levels has been discussed at lines 534-536.

Reviewer 3 Report

In the present study, Kamal et al. reported the proteomic signature of KRAS-induced secreted proteins that was confirmed in surgical tumour samples from resected PDAC patients. Furthermore, they validated the putative diagnostic performance of three candidates: Laminin-C2 (LAMC2), Tenascin-C (TNC), and Pentraxin-3 (PTX3), and investigated by ELISA quantification in two cohorts of PDAC patients (n = 200). The study is interesting and has a lot to provide for the discovery of pancreatic clinical biomarkers. However, it has several issues in terms of data presentation. Please see my comments below for further consideration.

  1. In the results section, the introduction of the data is not done properly. It is required to show the fold change of all the 6 clonal cell lines in terms of a heat map. The PCA plot should be shown to represent the variability among the clonal cell lines.
  2. Also, it is important to show the CVs of all the fold changes among the cell types in heavy and light labels.
  3. Plot the volcano plot to show the difference between heavy and light labels.
  4. In Figure 1C, it is not clear why the PPI network is important and what the rationale for preparing this graph. Please provide more details and an explanation of this figure. Line 258-260 describes the network with all the interaction types, which include homology and literature too. How much of this PPI interaction is confidence? In the string database, if you add any subset of proteins, it always shows the interaction. However, these are based on in silico analysis. So, it is important to use specific databases that are prepared using experimental results instead of computational analysis.
  5. Why do not all the clones have similar results? What is the reason for having the difference in the results among the same experimental clones?
  6. Line number 287-289 is the method section, there is no need to write anything in the results sections.
  7. Figure 2B representative images are very specific to tissue sections. Take the broader overview of all collective tissue and provide the image in the supplementary files. With this overall tissue image, collect the density plot for the intensity of the protein staining and plot the histogram adjacent to the representative’s main figures.
  8. No information was provided regarding the patient's KRAS mutation which was used for the plasma-based training and validation experiment. The same applies to histology and RT-PCR experiment. Because the beginning of the study starts with the identification of the KRAS G12V signature proteins and validation of these proteins. It is important for the study to report the sample type for KRAS mutation in validation experiments. Kindly report that patients selected for the study was Kras mutant positive or negative.

Author Response

Reply:

We thank the reviewer for her/his appreciation of our work.

Point 5. We would like to start from point 5 and highlight that the six cloned cell lines used for the proteomic analysis  are not simply replicates but are indeed six distinct cell lines showing heterogeneous features because of the transformation induced by the oncogene KRAS. This was specified in a sentence at lines 224-234.

“Next, to gain hints on the impact of KRAS-induced phenotype modifications, we immediately cloned the KRAS-HPDE cell line and selected different cloned lines for phenotype and functional characterization…”

The full description of these heterogeneous cloned lines is available in ref. 41 (Siddiqui et al, Oncoimmunology, 2017).             Briefly, 3 cell lines had an epithelial phenotype (E-cadherin positive), had low motility in an invasion assay in vitro and were unable to metastatize in vivo. Other three cell lines had a mesenchymal phenotype (E-cadherin low, Vimentin positive), had high Matrigel invasion ability and generated metastases in vivo. All the six cloned lines produced  inflammatory mediators (e.g. IL-6, CXCL8 and PTX3), however, a big distinction was that in the epithelial lines these inflammatory mediators were under the control of the master cytokine IL-1, while in the mesenchymal ones were not (this in essence was the message of the Oncoimmunology paper).

We felt that using heterogeneous KRAS-transduced pancreatic lines was a plus in our fishing expedition experiment in search of biomarkers.

Point 1. As suggested  for  the presentation of proteomic data, we have now added to Appendix A Supplemental Material new heatmaps with the fold change of the 163 dysregulated proteins  in all the 6 KRAS-cloned cell lines (Fig. S3). In addition, we have uploaded as separate files the heatmaps of the 437 non-dysregulated proteins (in alphabetical order) for all the 6 KRAS-cloned cell lines and Mock samples, as well as the global heatmaps of the 600 identified proteins, again for all the 6 KRAS-cloned cell lines and Mock samples (named as : Heatmaps of 437 non-dysregulated proteins in 6 KRAS-cloned cell lines and in 3 Mock samples; and Global heatmaps of all the 600 identified proteins in 6 KRAS-cloned cell lines and in 3 Mock samples).

The complete heatmaps will allow an interested reader to better evaluate the heterogeneity of the six KRAS-clones at the secretome level.  Moreover, the heatmaps may make clearer how the apparently chaotic KRAS-induced dysregulations only affect a portion of these secretomes, in contrast with the overall stability seen in the three Mock samples.

Point 2 and 3. In this study we put our effort in the discovery of secreted propteins for the potential clinical testing of early PDAC biomarkers. As the six cloned lines are not replicates of the same cell population but are indeed six distinct cell lines,  we did not focus on establishing the degree of similarity or diversity between the investigated secretomes. We thus reasoned that an in-depth statistical description of these highly heterogeneous secretome profiles was not of central interest in this study. 

Point 4. As to the uncertain relevance of Figure 1C (PPI network and interaction types) in the context of this paper, we agree with Reviewer 3 considerations, and have therefore removed Figure 1C.

Point 5: Already answered above.

Point 6. As suggested, the sentence has been removed from the Results section.

Point 7. As requested we have included immunohistochemistry pictures from all the 4 PDAC cases in the Supplementary file (Figure S5). We feel that a computer calculation of the stained area is not necessary in this type of study. The aim of this histology evalution was to demonstrate that in addition to mRNA expression for the proteins under study, we also detected protein expression. We do not discuss that protein expression in the tumor is associated to protein levels  in the circulation; we just wanted to be sure that proteins could be detected in tumor tissues. This was indeed the case for all the 4 tumors we tested.

Point 8.  The genotyping of KRAS mutation in PDAC tumors is not performed on a routine basis at our hospital (as in most hospitals). This analysis would require the re-collection of all the paraffin tumors, DNA extraction  and genotyping: quite challenging and very long to perform. We understand the concern of the reviewer, however, from the literature data, KRAS mutations have been demonstrated to occur in 95% of all patients, even at early stages, being usually the first appearing mutations.

Round 2

Reviewer 3 Report

No more comments.